# Shortcoming of the Mouse Model of Postoperative Ileus: Small Intestinal Lengths Have Similar Variations in In- and Outbred Mice and Cannot Be Predicted by Allometric Parameters

**DOI:** 10.3390/biomedicines13122948

**Published:** 2025-11-30

**Authors:** Maximiliane von Stumberg, Ejder Akinci, Berkan Ertim, Christina Oetzmann von Sochaczewski

**Affiliations:** Chirurgische Klinik, Universitätsklinikum Bonn, 53127 Bonn, Germany

**Keywords:** postoperative ileus, intestinal length, anatomical parameters, crown–rump length, body weight, mice, CD-1, C57Bl/6, age, variability

## Abstract

**Background/Objectives**: The mouse model of postoperative ileus separates the gastrointestinal tract into 15 sections, 10 of which are in the small intestine, to measure intestinal transit time. Usually, mice are standardised according to age or body weight. This inherently assumes that intestinal lengths are similar among the included mice irrespective of the method of standardisation. We aimed to test this assumption by comparing intestinal lengths, measuring their variability in commonly used out- and inbred strains. **Methods**: Mice were humanely killed, and their intestines were removed and measured in a standardised fashion. We compared the coefficients of variability via the modified signed-rank likelihood test. **Results**: We included 125 mice of the Crl:CD1(ICR) background and 10 mice of the C57Bl/6J and C57Bl/6NCrl substrains. The mean small intestinal length of Crl:CD1(ICR) mice was 437 mm (standard deviation 54), while it was 473 mm (standard deviation 29) in C57Bl/6J mice and 419 mm (standard deviation 57) in C57Bl/6NCrl mice. The respective coefficients of variation were 12.4%, 6.1%, and 13.6% and did not differ between the out- and inbred strains (modified signed likelihood ratio 5.878, *p* = 0.053). This was not the case for caecal and large intestinal lengths. **Conclusions**: Due to substantial variation in small intestinal length, the separation of the small intestine into ten equally sized segments to measure intestinal transit time might not be warranted. This could be addressed by measuring small intestinal transit time in absolute values and relative to the intestinal length.

## 1. Introduction

Postoperative ileus is a frequent complication of gastrointestinal surgery, affecting consistently more than 10% of prospectively studied cohorts irrespective of the healthcare system [1,2]. Its occurrence results in an increased length of hospital stay and higher hospital fees [3], negatively affecting outcomes for both patients and the healthcare system [4]. The definition of postoperative ileus consists of cessation of coordinated bowel movements after surgery, limiting intestinal transit and thereby oral intake of fluids and solid food [5]. Its complex pathophysiology with an interplay of inflammation, neural reflexes, and neurohumeral pathways [6,7] has prompted the investigation of different treatments, but these interventions are not equally beneficial for all patient populations. Knockout-mouse models causing different forms of immune disturbances in both the innate or adaptive immune system demonstrated increased severity of postoperative ileus [8,9,10,11]. Despite their influence on the pathogenesis, they have not been deemed suitable to determine whether a patient will experience a postoperative ileus [12]. Therefore, it is still difficult to identify those patients that will develop a postoperative ileus within the many who have abdominal surgery.

Among the many used outcome parameters in clinical studies is bowel transit time measured by radio-opaque markers [13]. This is also the parameter of choice in pre-clinical mouse models [14,15]. In them, intestinal transit is measured by the advancement of fluorescein-labelled dextran in the intestinal contents [15,16,17,18] or 99^m^-Technetium-labelled methylcellulose in earlier reports [19,20]. In order to compare intestinal transit, these models separated the small intestine into six equal segments [19] or the whole intestine into 15 segments: the stomach, 10 equally sized small intestinal segments, the caecum, and 3, sometimes 2 [21], equally sized large intestinal segments [15,16,17,18]. This approach has also been used in rats [22] and pigs [23], but was sometimes modified to 10 equally sized colonic segments based on the different anatomy of pigs [24]. Other seldom used approaches were a real-time measurement of gastrointestinal transit time using a SmartPill device [25,26] in pigs [27], phenol red in glucose [28], Evans Blue in sodium chloride [29], or charcoal in barium sulfate and sodium chloride [30].

These measurement approaches all inherently assume that the intestinal lengths are similar between the experimental units. This is further complicated by the fact that the standardisation of animals varies between studies: some employ a standardisation of mice by body weight [15,16,17,21,31,32,33], whereas others prefer age [18,19,34,35,36]. A combination of both is rarely favoured [37,38,39]. Nonetheless, this standardisation implies that intestinal lengths should be similar between the body weight or age ranges employed in the respective studies. Alternatively, one might also assume that the speed of propagation of the ingested aliment would be similar relative to the intestinal length. Moreover, several different mouse strains are used in the rodent models of postoperative ileus. The preferred mouse strain is C57Bl/6 [15,16,17,18,21,32,37,40,41], but BALB/c [19,20,42,43,44], C3H [45], hybrids [28], or Sprague Dawley rats [31,41,46] are also used in experiments.

This aspect represents a common issue in basic research, which often follows tradition in the choice of the model organism. This often results in using outbred rats, but inbred mice [47]. This is also the case for the model of postoperative ileus, which favours inbred mice, but outbred rats if rats are used. Historically, it had been assumed that inbred strains would be favourable due to their limited inter-individual variation based on their isogenicity and homozygosity [48,49]. However, newer research has shown that inbred strains might have a coefficient of variability of measurements similar to those of outbred strains [50]. To assess and quantify these effects, it has been suggested to use these strains in parallel [47], which is likely not to be achievable when using genetically modified organisms. They first have to be backcrossed to a background strain for ten generations to have a specific genetic background again [51]. However, it still allows the comparison of in- and outbred strains with regard to their variability.

We therefore aimed to compare the intestinal lengths and their variations in an outbred stock, Crl:CD1(ICR) mice, with two substrains of C57Bl/6, C57Bl/6J, and C57Bl/6N. Moreover, given the method of standardisation by age or body weight, we aimed to investigate whether these parameters would be suitable to estimate intestinal lengths.

## 2. Materials and Methods

We obtained mice from inhouse colonies of Crl:CD1(ICR), C57Bl/6J, and C57Bl/6NCrl, whose mice were bought from a vendor (Charles River, Sulzfeld, Germany). Mice were eligible for our study if they were either not included in a different study or if they were not used in them anymore. Thereby, we avoided killing a mouse exclusively for our study. All mice had already been at our facility for at least seven days [52,53] before inclusion in our experiment.

Mice were housed at the Haus für Experimentelle Therapie at our institution in a specific pathogen-free barrier in individually ventilated cages in groups of a maximum of five animals per cage. The cages were equipped with beddings made of dust-free European aspen in a chip size of two to three millimetres (Abedd Midi Chips, Abedd, Kalnciems, Latvia) and dust-free pulped cotton fibre nestlets (Ancare, Bellmore, NY, USA) as environmental enrichment. Cages were equipped with materials to play with in the form of gnawing sticks (Bricks M, Tapvei, Paekna, Estonia). Mice had autoclaved tap water available at libitum and were fed ad libitum with an appropriate pelleted regular chow (Ssniff V1534-300, Ssniff Spezialdiäten, Soest, Germany), which consisted of 67% carbohydrates, 24% protein, and 9% fat.

The husbandry barrier had a centrally regulated ventilation system that kept a constant room temperature of 22 degrees Celsius, a relative humidity of 45 to 65 percent, and ensured an air-exchange of 15 times per hour. We used a dark–light cycle of twelve hours, with artificial lighting between 7 o’clock and 19 o’clock. Mice were regularly checked according to standard protocols by keepers and by veterinarians if necessary. The cages were exchanged once weekly under aseptic conditions according to the standard procedures of our husbandry facility.

Mice were sacrificed in a different room of the facility in a type II long narcotic chamber (Tecniplast, Hohenpeißenberg, Germany) via gradually increasing carbon dioxide insufflation. Their demise was determined by a loss of respiratory excursions and ensured by cervical dislocation thereafter. We performed our experiments on multiple days with a different number of animals per day in order to increase the reproducibility of our experiments [54]. To reduce the time between killing and explanation of the gut from the mouse carcass, no more than five animals were sacrificed at the same time in order to avoid structural changes [55,56,57,58]. We measured body weight using an electric scale (Goldwell 1181, Kao, Darmstadt, Germany) with a resolution of 0.1 g and an accuracy of 0.01 g. Mice were placed in supine position and the necropsy was performed following standard protocols [59,60]: the peritoneal cavity was opened in the midline and the gut was cautiously removed en bloc after dissection at the oesophago–gastric junction and anorectal junction (Figure 1).

The different parts of the intestine were identified by anatomical landmarks. The small intestine was identified by the end of the pylorus and the ileocaecal valve, which also identified the caecum, whose transition into the large intestine was gauged from the tissue structure, leaving the large intestine. Afterwards, the segments were placed longitudinally without traction to ensure anatomically correct conditions. This was ensured by placing the intestinal segments along a ruler (Schneider & Baier, Heilbronn, Germany). Intestinal lengths were then measured via an electronic slide gauge (Kynup, Shenzen, China) with a resolution of 0.01 mm and an accuracy of 0.02 mm (Figure 2). Each part of the intestine was measured thrice, and the mean of these measurements was recorded as the value for the specimen. Measurements per mouse took around 20 min from the first to the last measurement.

Available data on intestinal lengths are sparse; these were only provided in the context of a by-product of different experiments, either from four mice of undisclosed C57Bl/6 background [61] or just three mice of a non-specified genetic background [62]. We therefore had to resort to the results of a preceding study, which calculated an adjusted *R*^2^ of 0.15 for oesophageal length [63]. Using the non-adjusted *R*^2^ of this study of 0.169, a conventional α = 0.05, and increased β = 0.9, we calculated the required number of experimental units as 104 to detect a difference in the *R*^2^ of the model to the null hypothesis of *R*^2^ = 0 for double-sided testing. The a priori required sample size was calculated using G*Power (version 3.1.9.7) [64].

Statistical analysis was conducted using GraphPad Prism 8 (Dotmatics, Boston, MA, USA). The analysis of the equality of coefficients of variation using the cvequality-package (version 0.2.0) [65] was conducted using R (version 4.3.3) [66]. Confidence intervals for the coefficients of variation and the considerations for sample size analysis using confidence intervals based on reviewer suggestions in the discussion were calculated using R’s MBESS-package (version 4.9.41) [67]. We assessed intestinal lengths for a Gaussian distribution using the Kolmogorov–Smirnov test supported by visual analysis of quantil–quantil plots [68,69,70]. This analysis was also applied to assess the normality of residuals in the regression analysis. Results of descriptive statistics were provided as mean and standard deviation. In addition, the variability of the different intestinal lengths was given as the coefficient of variation. Multivariable analyses were conducted using body weight, age, crown–rump length, and their interaction as independent variables to predict the different intestinal lengths. Multicollinearity was assessed via a correlation matrix of Pearson’s *R* for the respective independent variables and the variance inflation factors. The equality of the coefficients of variation was tested using the modified signed likelihood ratio test [71].

During peer-review, the reviewers suggested including exploratory analyses for the effect of the substantial age differences between the three included mouse strains. This was also conducted with R via an analysis of covariance for small intestinal length by strain with age as a covariate, using the rstatix-package (version 0.7.3) with post hoc testing via estimated marginal means [72].

Our experiments were compliant with directive 2010/63/EU, the national regulations for the protection of animals, and its statutory instruments. German law for the protection of animals exempts all experiments in which laboratory animals are sacrificed to obtain isolated organs from approval by the competent state authority (exact citation: section seven, subsection two, sentence three of the German law for the protection of animals [German legal citation: “Paragraph sieben, Absatz zwei, Satz drei des Tierschutzgesetzes”]) [63,73].

## 3. Results

We included 125 mice (*mus musculus*) of the Crl:CD1(ICR) outbred stock in our study, of which 8 were male and the remaining 117 were female. For comparison to inbred strains, we also included ten mice each of the C57Bl/6J and C57Bl/6NCrl substrains. Of the C57Bl/6J mice, six were male and four female, which was also the case for the C57Bl/6NCrl substrain.

The mean age of the Crl:CD1(ICR) mice was 98 days (standard deviation 27.2), while it was 268 days (standard deviation 72.6) for C57Bl/6J mice and 127 days (standard deviation 11.6) for the C57Bl/6NCrl mice. With regard to body weight, the mean body weight for Crl:CD1(ICR) mice was 36.3 g (standard deviation 6.2), while it was 33.1 g (standard deviation 2.7) for C57Bl/6J mice and 27.1 g (standard deviation 2.9) for C57Bl/6NCrl mice. Mean crown–rump length of Crl:CD1(ICR) mice was 73.7 cm (standard deviation 9.3), whereas it was 71.9 cm (standard deviation 4.5) for C57Bl/6J and 62.6 cm (standard deviation 2.9) for C57Bl/6NCrl.

The mean small intestinal length of Crl:CD1(ICR) mice was 437 mm (standard deviation 54), while it was 473 mm (standard deviation 29) in C57Bl/6J mice and 419 mm (standard deviation 57) in C57Bl/6NCrl mice (Figure 3). The respective coefficients of variation were 12.4% (95% confidence interval: 11–14.2%), 6.1% (95% confidence interval: 4.2–11.2%), and 13.6% (95% confidence interval: 9.3–25.2%). They were more likely to be similar between the out- and inbred strains (modified signed likelihood ratio 5.878, *p* = 0.053).

Mean caecal lengths were 35.3 mm (standard deviation 6.1) for Crl:CD1(ICR) mice, 32 mm (standard deviation 2) for C57Bl/6J mice, and 31.4 mm (standard deviation 3.1) for C57Bl/6NCrl mice (Figure 4). Their respective coefficients of variation were 17.4% (95% confidence interval: 15.4–20%), 6.1% (95% confidence interval: 4.2–11.1%), and 9.7% (95% confidence intervals: 6.8–17.9%). These coefficients were not equal between them (modified signed likelihood ratio 14.07, *p* < 0.001).

The mean large intestinal length of Crl:CD1(ICR) mice was 102 mm (standard deviation 20), while those of C57Bl/6J mice was 88.7 mm (standard deviation 9.4) and those of C57Bl/6NCrl was 76.9 mm (standard deviation 6.1) (Figure 5). The coefficients of variation were 20% (95% confidence interval: 17.7–23%) for Crl:CD1(ICR) mice, 10.6% (95% confidence interval: 7.4–19.5%) for C57Bl/6J, and 7.9% (95% confidence interval: 5.5–14.5%) for C57Bl/6NCrl. These coefficients of variation were also not equal between the included strains (modified signed likelihood ratio 12.98, *p* = 0.002).

Before the regression analysis for Crl:CD1(ICR) mice, we investigated whether there was substantial correlation between the included independent variables age, crown–rump length, and body weight. This was indeed the case between body weight and crown–rump length with a *R* of 0.7 (95% confidence interval 0.6–0.78), whereas the correlation coefficients involving age were slightly smaller with *R* = 0.67 (95% confidence interval 0.56–0.76) between body weight and age, and *R* = 0.62 (95% confidence interval 0.5–0.72) between crown-rump length and age (Figure 6).

We also evaluated whether correlations were present between the different dependent variables of small intestinal length, caecal length, and large intestinal length. Relevant correlations between dependent variables could only be found between small and large bowel length with *R* = 0.55 (95% confidence interval 0.42–0.66) (Figure 7).

The planned regression analysis was precluded by the substantial multicollinearity between the independent predictors. This can be exemplified by variance inflation factors of 37 for crown–rump length, 113 for body weight, and 174 for age.

In order to account for potential effects of the different age of the mice between the different strains, we conducted an exploratory analysis of the effect of age on small intestinal length using an analysis of covariance. It indicated a statistically significant difference for age (*F*(1,141) = 12.03, *p* < 0.001), but not for the strain (*F*(2,141) = 1.56, *p* = 0.22). However, the post hoc pairwise testing did not show any differences between the strains adjusted for the covariate of age (*p* ≥ 0.12).

## 4. Discussion

Postoperative ileus is a major complication of abdominal surgery, with relevant negative impacts for both the healthcare system and the individual patient [4]. The workhorse for experimental investigations in postoperative ileus is the murine model of postoperative ileus [14,15]. It offers the advantage of a small animal with short generation time and the opportunity to directly evaluate potential mechanisms. The latter is performed by using genetically modified organisms such as inborn knockouts or induced ones by, for example, Cre-LoxP [74]. The finding of postoperative ileus manifesting itself in the stomach and small intestine in the murine model had resulted in the focus on intestinal transit time as a relevant measure of effect of potential therapeutic and prophylactic options [75].

Currently, mice used in experiments for postoperative ileus are standardised by body weight [15,16,21,31,32] or age [18,19,34,35,36]. This approach to standardisation within the experiment inherently assumes that both parameters are at least associated with intestinal lengths. Due to a lack of data on that matter, it seemed reasonable to test this assumption and elucidate which parameter, body weight or age, would have a higher influence on intestinal length. This could have resulted in a higher level of standardisation with subsequently less variation. Thus, researchers could potentially have detected effects with a lower effect size. Notably, our results do not question any of the identified modifying factors in experimental postoperative ileus. These factors had effect sizes that were sufficiently large to be detected in a cohort with a higher level of variation, which would not have changed in a cohort with less variation. Although heterogenisation has been advocated for to avoid the standardisation fallacy [76], this does not suggest that variation should not be reduced. This concept deals with known sources of residual variation [77], but does not oppose standardisation to avoid excessive variation from unknown sources.

It would have not been surprising if body weight had been an influential factor for intestinal length. Mice grow throughout their life [78] and organ weights increase in parallel to body weight in rats [79], but age in mice [80]. These differences between the two rodent species highlight the uncertainty in these matters. Interestingly, crown–rump length is usually not addressed in experimental studies, although it has been described to be the only influential factor for small bowel length in children [81] and adults [82]. Consequently, we included crown–rump length into our evaluation of potentially predictive factors for intestinal lengths. Due to the potential interplay between the predictors, we included pre-planned two-way interactions into our regression analyses. That this assumption was warranted can be demonstrated by the substantial correlation between the predictor variables body weight, age, and crown–rump length in our study. Their correlation resulted in substantial multicollinearity that precluded the intended regression analysis.

Our study could therefore not achieve its intended aim, which was to investigate which allometric parameter would be most suitable to standardise mice for studies using the mouse model of postoperative ileus [14,15]. Nevertheless, our study highlights an important shortcoming of the current mouse model of postoperative ileus. In its present form with 10 equally sized small intestinal parts, an assumption of the model is that small intestinal lengths of the included mice are similar. However, our results indicate that this is likely not the case. We were able to show that the coefficient of variation in small intestinal length was similar between the included in- and outbred mice. The coefficient of variation shows the extent of variability in relation to the mean of the investigated population. It is a commonly used parameter to address variability in experimental animals [50,83,84]. For our research question, the mean intestinal length was of lesser relevance, but the potential variation was the measure of choice as this is part of the variability in the mouse model of postoperative ileus.

One might argue that we compared inadequately sampled populations from the Crl:CD1(ICR), the C57Bl/6J, and C57Bl/6NCrl mice due to their increased age. Due to our adherence to the 3R-guidelines and the ethical obligations arising from it, we were not free to decide their ages as we had to take them when they were no longer needed for other experiments. Although the ages of the different genetic backgrounds differ, this is unlikely to have a relevant effect for the matter of similar coefficients of variation in small intestinal lengths. The explanation for this is rooted in the growth curve of mice [78,85,86]. While the growth in body weight is linear to age in days in the first days of life [78,85], it speeds up afterwards to almost exponential growth [87]. These growth curves were also applicable to the C57Bl/6 background [88] and ICR-outbred [89] mice. Of note, the variability in body weight of the C57Bl/6 background has been described to be rather small compared to other strains [90]. So, for example, standardisation by age often includes mice between six to nine weeks [18], six to ten weeks [19], or eight to twelve weeks [91], while they were only scarcely limited to “~6 weeks” [34]. As these age intervals are still within the parts of the growth curves where substantial weight gain occurs, this standardisation is likely to introduce substantial variation. This is also the case for 20–25 g [16,17,21], even more for 12–16 g [46], and also for “approximately 25 g” [33] or “~25 g” [32] due to the associated positions on the growth curves. This suggests that the variability in mice sampled from these parts of the growth curves is likely to be even higher than in our cohorts, where additional weight gains are much smaller. The analysis of covariance with subsequent post hoc testing included in our study during peer-review did not support an effect of age on intestinal length.

Another issue and limitation that pertains to the sampled population is the uneven distribution of sexes in our cohort. We did not impose a constraint regarding the sex of the included animals based on the widely described sex-bias in experimental research, varying by research area [92]. Instead we aimed to include both sexes following recommendations to reduce this bias [93]. Consequently, we included mice as they were available following our approach as they were not selected for other experiments. Regarding the C57Bl/6 substrains, this resulted in a somewhat balanced distribution of sexes, but the Crl:CD1(ICR) mice were predominantly female. This predominance was to such an extent, more than 90% females, that it precluded the exploration of sex as a covariate. Moreover, the female predominance in the Crl:CD1(ICR) group can therefore only be considered as a limitation, because it could be the case that there are sex-specific differences that influenced our results.

Our study also highlights that one cannot evade the problem of variability by using inbred mice strains. It has been shown in a meta-study that the coefficients of variation and thus the variability often does not differ between in- and outbred strains [50]. Although C57Bl/6 substrains often exhibit less variability than other strains in anatomy [94] and physiology [83], these differences in variability are seldom of relevance, which makes outbred strains equally suitable in many experimental designs. The smaller variability in the C57Bl/6J substrain could also be demonstrated in our results as the coefficient of variability of small intestinal lengths was more than halved compared to the Crl:CD1(ICR) outbred stock. Likewise, the small intestinal length also demonstrated that coefficients of variability might be numerically higher in in- than outbred strains, as that of the C57Bl/6NCrl substrain was higher than that of the Crl:CD-1(ICR) strain.

An additional limitation of our study is that we did not perform an a priori power analysis for the comparison of the coefficients of variation. However, we did conduct a pre-planned power analysis based on data from the literature, with a meaningful outcome for the regression analysis. This sample size analysis prompted us to include more than 100 mice alone from the Crl:CD-1(ICR) background. Given these substantial numbers of included mice, it is unlikely that including additional mice of this background would have changed the results of the comparison. This is also the case for the number of inbred mice. The coefficients of variation in small intestinal length and caecal length of the C57Bl/6J substrain were already the smallest in our study.

In addition, the only available data from the literature to calculate a coefficient of variation from just three C57Bl/6J mice reports a coefficient of variation of 7.6% for the small intestinal length if just the standard deviation of the jejunum is used [61]. This coefficient of variation is even larger if the variances of the different small intestinal parts are summed up, where the coefficient of variation would be 8.5%. Let us assume that we define a coefficient of variation that is doubled in the outbred strain compared to the C57Bl/6 substrains as relevant and further assume that the small intestinal lengths are similar based on the results of Michel et al. [61]. With 125 mice, the confidence interval of a doubled coefficient of variation, 15.2%, could be narrowed to between 13.1% and 17.3% based on Kelley’s method [95], with the usual degree of certainty being 80% as is common in power analyses. For the 10 mice of the C57Bl/6 substrains, the respective confidence interval around the coefficient of variation of 7.6% would range between 2.1% and 13.1%, indicating that a relevant difference in coefficients of variation could be detected using our sample sizes. This indicates that our sample size would have been sufficient to detect a relevant difference in the coefficients of variation between in- and outbred strains. However, it has to be viewed with caution as this was calculated after the fact based on reviewer suggestion and therefore does not have the same value as a priori sample size calculation. We refrained from calculating post hoc power of our study as it is a flawed concept because it is essentially a direct transformation of the *p*-value [96].

Another limitation of our study is that it focused purely on an anatomic parameter, the intestinal length. Obviously, one cannot deduce functional from structural parameters and our study did not assess functional outcomes. It could be the case that the peristalsis and subsequently the speed of chyme propagation were related to intestinal length. However, this is merely speculation and requires further investigation in functional studies as our study only addressed a structural parameter.

The shortcoming identified by our study may be addressed by measuring the absolute intestinal transit distance [97,98], and for comparability between substantially different intestinal lengths the relative intestinal transit as the ratio of the intestinal transit and the absolute small intestinal length could be used [98]. Using these approaches in parallel to the current technique of segmentation of the small intestine into ten equally sized segments would allow a comparison between them and determine which of the two methods would be more appropriate.

Taken together, we were able to show that the variability of small intestinal lengths is similar between in- and outbred mice and present to a substantial account. Our results therefore do not support the concept of measuring intestinal transit via the propagation along ten equally sized small intestinal segments based on the substantial variability of small intestinal length as an anatomic parameter. Choosing to do so may result in missing potentially successful interventions due to smaller effect sizes. Standardisation by body weight or age is unlikely to change this finding.

## Figures and Tables

**Figure 1 biomedicines-13-02948-f001:**
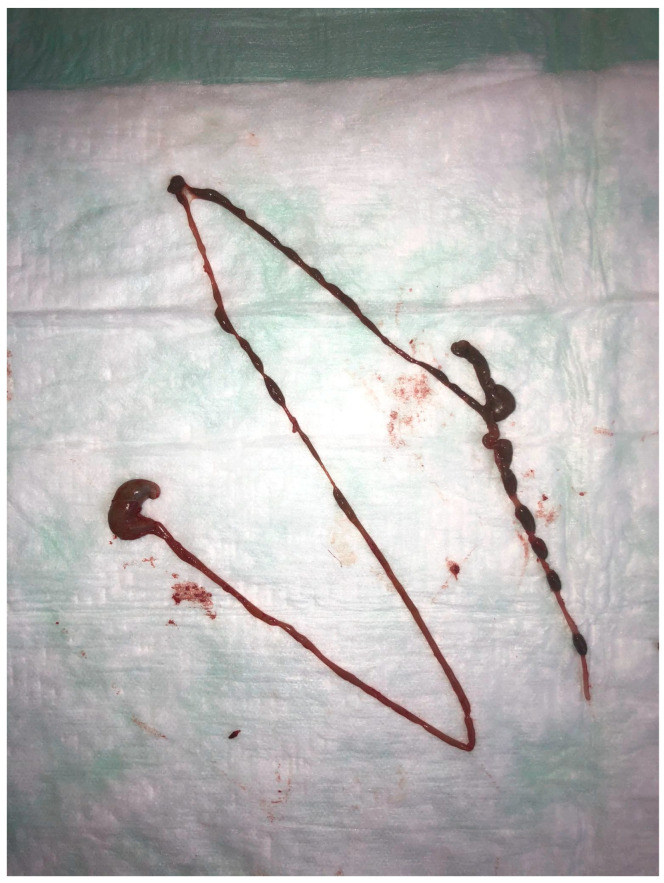
The whole intestinal organs en bloc after being freed from surrounding tissue and dissected cranially at the oesophago–gastric and caudally at the anorectal junction.

**Figure 2 biomedicines-13-02948-f002:**
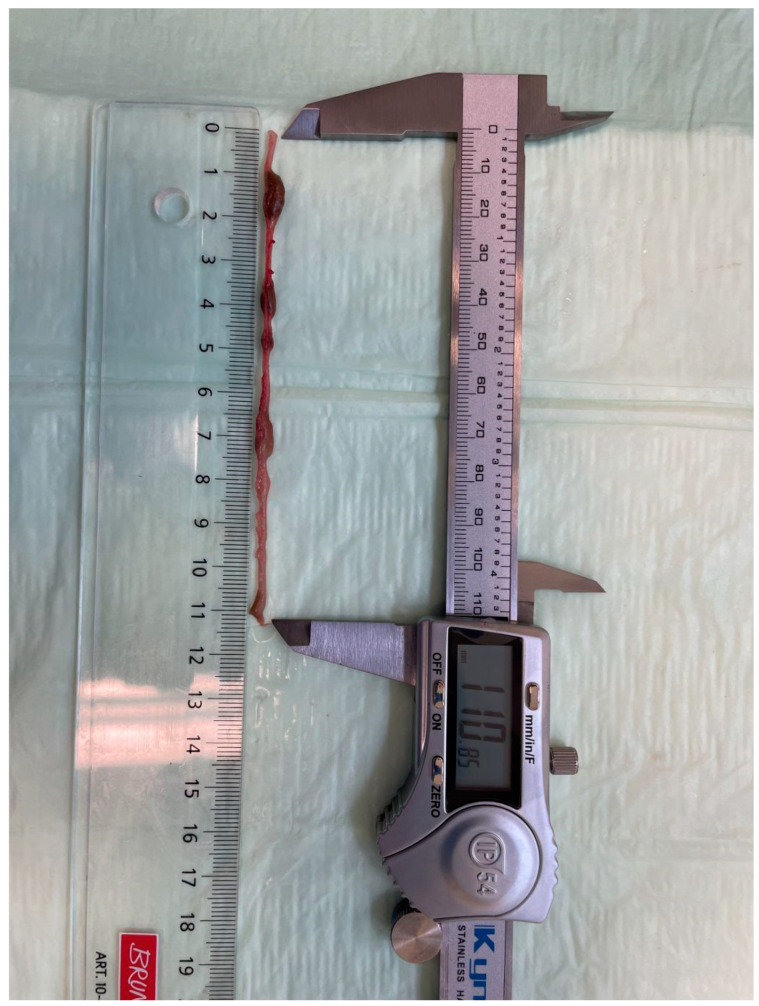
Measuring the intestinal segments via the electronic slide gauge in a straightened position ensured by placing them along a ruler.

**Figure 3 biomedicines-13-02948-f003:**
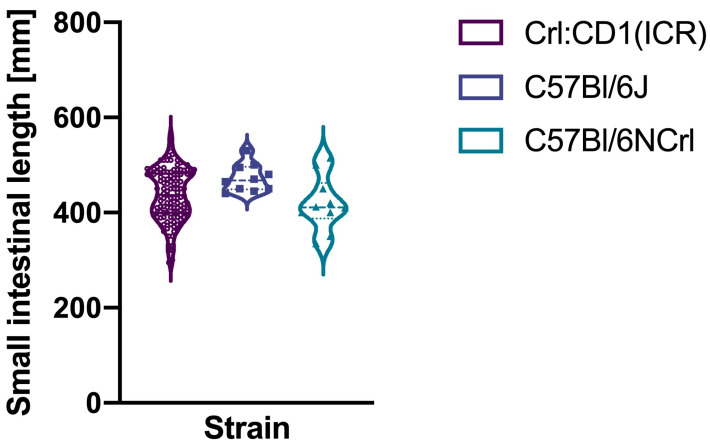
Small intestinal lengths of the three mice strains.

**Figure 4 biomedicines-13-02948-f004:**
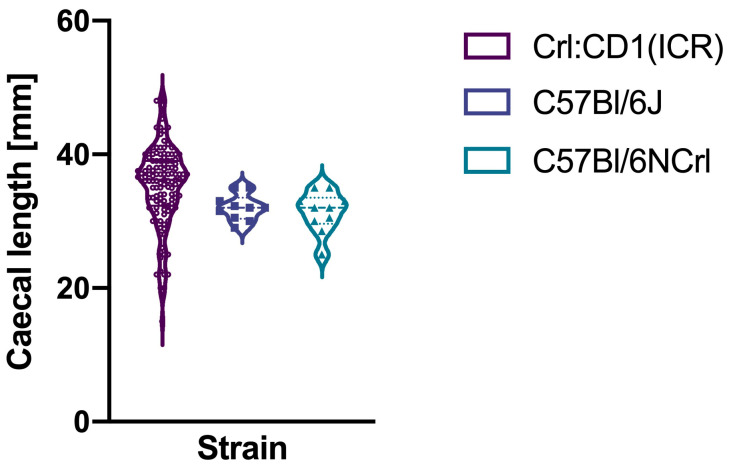
Caecal lengths of the three mice strains.

**Figure 5 biomedicines-13-02948-f005:**
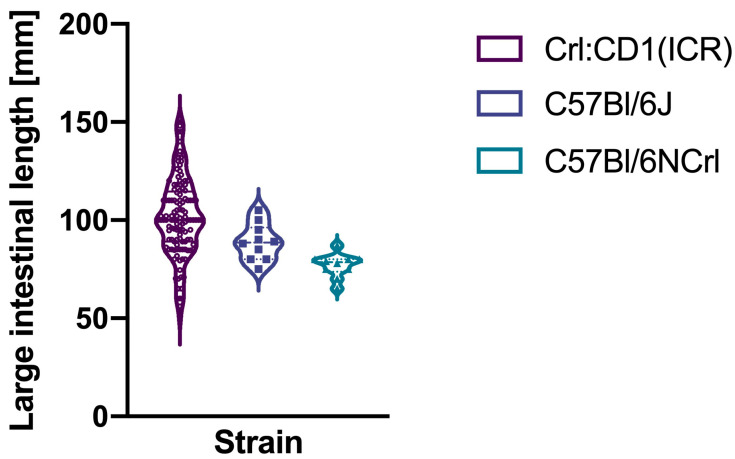
Large intestinal lengths of the three mice strains.

**Figure 6 biomedicines-13-02948-f006:**
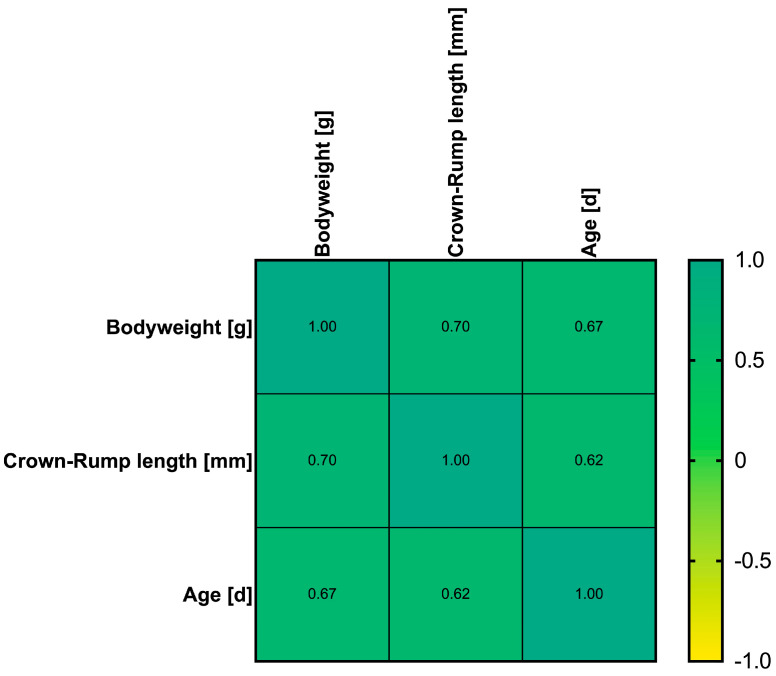
Correlogram of correlation coefficients between the independent predictors age (in days), crown–rump length (in millimetres), and body weight (in grams) in Crl:CD1(ICR) mice.

**Figure 7 biomedicines-13-02948-f007:**
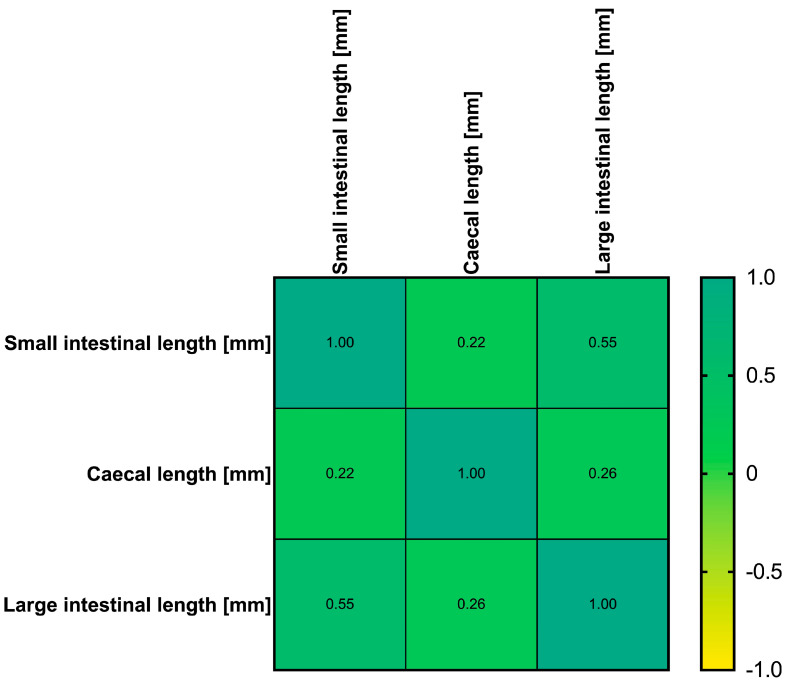
Correlogram of correlation coefficients between the dependent variables small bowel length (in millimetres), caecal length (in millimetres), and large bowel length (in millimetres) in Crl:CD1(ICR) mice.

## Data Availability

Raw data of our study are freely available from *Zenodo* (https://dx.doi.org/10.5281/zenodo.17486190) (access date 30 November 2025).

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
