# Peer review of "Shortcoming of the Mouse Model of Postoperative Ileus: Small Intestinal Lengths Have Similar Variations in In- and Outbred Mice and Cannot Be Predicted by Allometric Parameters"

_biomedicines, 2025, doi:10.3390/biomedicines13122948_

Round 1

Reviewer 1 Report

Comments and Suggestions for Authors

The authors suggest that current methods to standardize the digestive tract in post op ileum are not completely reliable as the intestinal length can not be precisely estimated by using the age and weight of the animal. Refinement of animal models is essential to optimize scientific outcomes and although the authors do not propose an alternative to the current method used, providing information on the reliability of it is also useful. 

The number of animals, sex and ages used were determined by the availability of animals used for other purposes, which is of value. CD1 end up used in a group of ages where obesity is not an issue. However, a better representation between males and females should have been considered.

B6 animals were however quite old and there is a concern if results related to weight could be affected by the body weight and if body weigh in these animals was overestimated as they were old. The authors should address this issue or reformulate additional groups that are representative of different ages and weights

Author Response

Reviewer 1:

The authors suggest that current methods to standardize the digestive tract in post op ileum are not completely reliable as the intestinal length can not be precisely estimated by using the age and weight of the animal. Refinement of animal models is essential to optimize scientific outcomes and although the authors do not propose an alternative to the current method used, providing information on the reliability of it is also useful. 

We thank the reviewer for considering our research as useful to improve the animal model. Of course, we could not propose a different approach. We have however included a discussion of approaches available in the literature to mitigate this potential shortcoming. We have marked all the changes in our manuscript following your recommendations in green. As you suggested that we review the use of the English language again, we had a colleague who speaks English as a first language review the manuscript and implemented several changes to improve the use of the English language.

The number of animals, sex and ages used were determined by the availability of animals used for other purposes, which is of value. CD1 end up used in a group of ages where obesity is not an issue. However, a better representation between males and females should have been considered.

We agree that a better representation of the two sexes would have been favourable. However, following the recommendations to not impose sex restrictions in research, we included both sexes and had no influence on the supplied sexes. We have included this aspect in the discussion among the limitations of our study following the paragraph discussing the sampling of the cohorts.

B6 animals were however quite old and there is a concern if results related to weight could be affected by the body weight and if body weigh in these animals was overestimated as they were old. The authors should address this issue or reformulate additional groups that are representative of different ages and weights.

Reviewer 2 also suggested to better address this aspect. We therefore conducted an exploratory analysis of covariance taking into account age. Although the analysis of covariance was statistically significant, we did not found any differences between the strains in the post-hoc analysis adjusted for age.

Reviewer 2 Report

Comments and Suggestions for Authors

The manuscript entitled “Shortcoming of the mouse model of postoperative ileus: Small intestinal lengths have a similar variation in in- and outbred mice and cannot be predicted by allometric parameters” is an interesting topic. However, there are still several parts that need to be revised:

  1. There is a significant age difference among the samples, which may affect the comparison of variations in intestinal length. Although the author points out that age differences have little effect on the coefficient of variation, the average age of C57Bl/6J mice is 268 days, much higher than CD-1 (98 days) and C57Bl/6NCrl (127 days). It is suggested that the author further analyze the potential impact of age on small intestine length in the discussion, or consider using age as a covariate for stricter statistical control.
  2. The author acknowledges that no efficacy analysis was conducted on the sample size for comparing coefficients of variation, which may affect the reliability of the results. Suggest supplementing post hoc efficacy analysis or explaining whether the existing sample size is sufficient to detect significant differences in coefficients of variation, in order to enhance the statistical validity of research conclusions.
  3. The author points out that the current model assumes that the length of the small intestine is similar, and therefore proposes that the 10 equal division method should not be used to measure intestinal peristalsis time. However, this article only provides evidence of length variation and does not directly verify the impact of this segmentation method on experimental results. It is suggested that future research provide more direct evidence to support its viewpoint.
  4. This article focuses on anatomical parameters (length), but does not address intestinal function (such as peristaltic velocity, transport time, etc.). It is suggested that the author clearly point out the limitations of this study in the discussion, namely the failure to evaluate whether intestinal function is related to length variation, in order to avoid excessive inference of its impact on the effectiveness of the experimental model.
  5. The introduction and discussion sections are quite lengthy, and some literature reviews have weak relevance to the research question of this article. Suggest simplifying the background introduction, highlighting the research gaps and contributions of this article, and improving the logical compactness and readability of the article.

Author Response

Reviewer 2:

The manuscript entitled “Shortcoming of the mouse model of postoperative ileus: Small intestinal lengths have a similar variation in in- and outbred mice and cannot be predicted by allometric parameters” is an interesting topic. However, there are still several parts that need to be revised:

  • There is a significant age difference among the samples, which may affect the comparison of variations in intestinal length. Although the author points out that age differences have little effect on the coefficient of variation, the average age of C57Bl/6J mice is 268 days, much higher than CD-1 (98 days) and C57Bl/6NCrl (127 days). It is suggested that the author further analyze the potential impact of age on small intestine length in the discussion, or consider using age as a covariate for stricter statistical control.

We thank the reviewer for his recommendations to improve the strength and focus of our study. All changes made in regard to his/her request are marked in yellow. As Reviewer 1 has also pointed out this issue and suggested to further address this point using a post-hoc analysis, we have included an analysis of covariance using age and bodyweight to further explore this issue. Although the analysis of covariance was statistically significant, we did not found any differences between the strains in the post-hoc analysis adjusted for age.

  • The author acknowledges that no efficacy analysis was conducted on the sample size for comparing coefficients of variation, which may affect the reliability of the results. Suggest supplementing post hoc efficacy analysis or explaining whether the existing sample size is sufficient to detect significant differences in coefficients of variation, in order to enhance the statistical validity of research conclusions.

We chose to explain that the sample size would be sufficient given the small confidence interval that could be achieved using the available number of mice in the Crl:CD1(ICR) group. In general, analyses using the coefficient of variation are sparse and the authors are not aware of a method to estimate sample size apart from the one proposed by Kelley[1]. We have included the approach by Kelley used to calculate the confidence interval in the discussion and have also included the work by Michel et al.[2], who show that the coefficient of variability of C57Bl/6 being six weeks old is higher than the one encountered in our study. Using Kelley’s method, one could calculate a confidence interval ranging from 13.1% to 17.3% around the coefficient of variation of 15.2% (double the coefficient of variation from Michel et al.) with n=125 mice that would not cover the confidence interval of a coefficient of variation of 7.6% with n=10 resulting in an interval from 2.1% to 13.1% under the assumption that the small bowel length would be equal. The equality of small bowel lengths is one of the assumptions of the model of postoperative ileus, so this assumption seems valid. Of note, this is a conservative approach using the highest reported standard deviation of 2.29 for the jejunum from Michel et al. Even when the conservative approach using the standard deviation of 2.29 was tested in the modified signed likelihood ratio test with 10,000 simulations, it returned a highly statistically significant result with an MLRS of 10.82 and a p-value of 0.0045. We have implemented this aspect into the discussion using the line of argument based on the confidence intervals calculated by Kelley’s method.

  • The author points out that the current model assumes that the length of the small intestine is similar, and therefore proposes that the 10 equal division method should not be used to measure intestinal peristalsis time. However, this article only provides evidence of length variation and does not directly verify the impact of this segmentation method on experimental results. It is suggested that future research provide more direct evidence to support its viewpoint.

This is an important issue that you raised. Of course, we could not address the issue of function in our study aimed to assess the anatomical variation. We have however included a suggestion how this can be addressed in studies by measuring the intestinal transit time with regard to the conventional 10 segments and the relative and absolute intestinal length in parallel, which would further elucidate this aspect.

  • This article focuses on anatomical parameters (length), but does not address intestinal function (such as peristaltic velocity, transport time, etc.). It is suggested that the author clearly point out the limitations of this study in the discussion, namely the failure to evaluate whether intestinal function is related to length variation, in order to avoid excessive inference of its impact on the effectiveness of the experimental model.

Thank you for pointing out that important point. We have included a caveat in the discussion to better emphasize the limitation of our study that uses only structural aspects, but has not assessed functional aspects. We therefore have clarified this limitation in the discussion and pointed out that our results cannot be used to deduce any points regarding the functional outcome.

  • The introduction and discussion sections are quite lengthy, and some literature reviews have weak relevance to the research question of this article. Suggest simplifying the background introduction, highlighting the research gaps and contributions of this article, and improving the logical compactness and readability of the article.

During the write-up of our study, we were unsure of the extent of the introduction and the discussion, because our study addressed both aspects of laboratory animal science and the experimental model of postoperative ileus. We are happy to being given the opportunity to shorten our manuscript to make it more concise. Despite the addition of five new paragraphs after peer-review, the word count of our article increased by only 5 words. In addition, the number of cited literature has been reduced to 98 citations from 122.

References

  1. Kelley, K. Sample Size Planning for the Coefficient of Variation from the Accuracy in Parameter Estimation Approach. Behavior Research Methods 2007, 39, 755–766, doi:10.3758/BF03192966.
  2. Michel, K.; Kuch, B.; Dengler, S.; Demir, I.E.; Zeller, F.; Schemann, M. How Big Is the Little Brain in the Gut? Neuronal Numbers in the Enteric Nervous System of Mice, Guinea Pig, and Human. Neurogastroenterol Motil 2022, 34, e14440, doi:10.1111/nmo.14440.